# Exploring the Biocontrol Capability of Non-Mycotoxigenic Strains of *Penicillium expansum*

**DOI:** 10.3390/toxins16010052

**Published:** 2024-01-17

**Authors:** Belén Llobregat, Luis González-Candelas, Ana-Rosa Ballester

**Affiliations:** Instituto de Agroquímica y Tecnología de Alimentos (IATA), CSIC, Calle Catedrático Agustín Escardino 7, 46980 Paterna, Spain; belen.llobregat@iata.csic.es (B.L.); lgonzalez@iata.csic.es (L.G.-C.)

**Keywords:** competition, mycotoxin, patulin, secondary metabolism, VELVET complex, knockout, biocontrol

## Abstract

*Penicillium expansum* is one the major postharvest pathogens of pome fruit during postharvest handling and storage. This fungus also produces patulin, which is a highly toxic mycotoxin that can contaminate infected fruits and their derived products and whose levels are regulated in many countries. In this study, we investigated the biocontrol potential of non-mycotoxigenic strains of *Penicillium expansum* against a mycotoxigenic strain. We analyzed the competitive behavior of two knockout mutants that were unable to produce patulin. The first mutant (∆*patK*) involved the deletion of the *patK* gene, which is the initial gene in patulin biosynthesis. The second mutant (∆*veA*) involved the deletion of *veA*, which is a global regulator of primary and secondary metabolism. At the phenotypic level, the ∆*patK* mutant exhibited similar phenotypic characteristics to the wild-type strain. In contrast, the ∆*veA* mutant displayed altered growth characteristics compared with the wild type, including reduced conidiation and abnormal conidiophores. Neither mutant produced patulin under the tested conditions. Under various stress conditions, the ∆*veA* mutants exhibited reduced growth and conidiation when exposed to stressors, including cell membrane stress, oxidative stress, osmotic stress, and different pH values. However, no significant changes were observed in the ∆*patK* mutant. In competitive growth experiments, the presence of non-mycotoxigenic strains reduced the population of the wild-type strain during in vitro growth. Furthermore, the addition of either of the non-mycotoxigenic strains resulted in a significant decrease in patulin levels. Overall, our results suggest the potential use of non-mycotoxigenic mutants, particularly ∆*patK* mutants, as biocontrol agents to reduce patulin contamination in food and feed.

## 1. Introduction

*Penicillium* is a well-known genus of filamentous ascomycete fungi and is taxonomically classified within the family Aspergillaceae. While primarily found in soil, it has also been identified in grains, seeds, and numerous foodstuffs, including decaying organic materials. As of now, there are 483 recognized species within this genus [1]. These fungi are quite prevalent and have a variety of uses, playing essential roles in the biotechnology, pharmaceutical, and food sectors. Therefore, they are of great economic importance [1]. Most *Penicillium* species are characterized by their saprophytic behavior. However, there are a limited number of pathogenic species, such as *Penicillium digitatum*, *Penicillium expansum*, and *Penicillium italicum*, which can invade fruits, such as apples and oranges [2,3]. Secondary metabolites (SMs), which are widely produced by this genus of fungal species, are physiologically active substances that vary from powerful pharmaceuticals to mycotoxins that are toxic to both humans and animals [4,5].

Mycotoxins pose a risk to food safety and food security, causing substantial economic losses worldwide annually. Another negative economic impact of mycotoxin contamination is the loss of agricultural products, animal feed, and animal products once mycotoxins are detected [6]. *P. expansum* can enter fruit tissues through wounds during harvest, postharvest, and storage, causing maceration and decay [2]. Small rounded lesions quickly develop into soft rotting tissue. A pale-colored mold forms at the site of fungus penetration, which is typically located in the center of the rotting area, and later adopts the distinctive green-blue shade of conidia. Therefore, as these conidia are dispersed by wind, water, and insects to other fruits, they may cause more diseases. Infrequently, fungi may enter the pedicel. In such cases, deterioration occurs internally and does not manifest any symptoms on the fruit’s surface. Consequently, these fruits are not discarded and may become contaminated with mycotoxins [7]. While apples are the primary host of *P. expansum*, this fungal pathogen has also been found in pears, cherries, peaches, plums, walnuts, pecans, hazelnuts, and acorns [7,8]. *P. expansum* is the causal agent of the blue mold disease in pome fruit and is the main producer of patulin [9]. It is characterized as a heat-resistant lactone with a molecular weight of 154.12 g/mol and a melting point of 110 °C. The usual route of mycotoxin biosynthesis consists of an enzymatic cascade. In this enzymatic cascade, enzymes are sequentially activated by the new product generated upstream. In addition, these enzymes that catalyze multiple steps in the biosynthetic pathways often form gene associations, also known as biosynthetic clusters. The biosynthetic gene cluster of patulin in *P. expansum* consists of 15 genes [10,11,12]. Among these genes, three encode transporters (PatA, PatC, and PatM), one encodes a putative transcription factor (PatL), two have an unknown function (PatF and PatJ), and nine encode biosynthetic enzymes. The first enzyme of the pathway involved in patulin production is PatK, which is a 6-methylsalicylic acid synthase (6-MSAS) belonging to the group of type I polyketide synthases (PKSs). On the other hand, global transcription factors, which are encoded by genes outside biosynthetic gene clusters, are known to play crucial roles in regulating growth, morphological development, and the biosynthesis of secondary metabolites. These genes can be influenced by various environmental factors, including oxidative stress, pH, temperature, nitrate and iron availability, carbon sources, and light. All these factors can impact their expression. One such transcription factor is VeA, a phosphoprotein member of the VELVET family, which also includes VelB, VosA, and VelC [13,14]. VeA is unique to filamentous fungi and possesses the “VELVET domain”, which is a characteristic shared by all members of the VELVET protein family. VeA is involved in many cellular processes, including the response to oxidative stress, light-dependent control of sexual or sclerotial development, asexual or conidial development, morphogenesis, secondary metabolism, and, on occasion, virulence [13,14,15,16].

Patulin synthesis can be influenced by various physiological factors, including temperature and pH [7]; specifically, patulin secretion occurs at ambient pH (3–5), whereas the production is negatively affected by higher pH. *P. expansum* is a psychrotolerant fungus capable of growth even at extremely low temperatures, with an optimum growth temperature of approximately 24 °C. This poses a challenge for prevention because the fungus thrives and can produce patulin under the temperature conditions typically used for the cold storage of fruits. Consequently, preventing its growth and patulin production is difficult. Patulin cannot be detected by taste or smell and is found in both apples and apple-based products. It is classified as a mycotoxin with adverse effects on human health [17]. Specifically, like other mycotoxins, patulin can suppress the immune system [18]. Acute exposure to patulin can result in gastrointestinal symptoms, such as nausea, vomiting, ulcers, intestinal hemorrhage, and duodenal damage. Although there is insufficient data from animal and epidemiological research to conclusively demonstrate the carcinogenicity of patulin, the International Agency for Research on Cancer (I.A.R.C.) has classified it as a group 3 possible carcinogen. This classification was made despite evidence of genotoxicity observed in vitro and cell culture studies [19,20]. To address this concern, the maximum level of patulin in apple juice and apple cider has been limited to no more than 50 g/L by the European Commission, United States Food and Drug Administration, Ministry of Health of the People’s Republic of China, and Health Canada. According to the European Commission, stricter regulations are in place for products intended for infants and young children, with limits of 25 and 10 g/kg, respectively, for solid apple products, such as apple sauce. The optimal strategy for minimizing mycotoxin levels in the food chain involves preventing fungal growth. In cases where the fungus has already developed, efforts should be made to inhibit the release of toxins. Several methods were suggested, including chemical interventions, physical controls, and biological treatments [1]. Specifically, the use of antagonistic microorganisms is a crucial management technique for fungal diseases. Biocontrol microorganisms employ four basic strategies: competition for resources and accessible spaces, antibiotic production, induction of resistance, and direct parasitism.

Competitive exclusion is the primary method used for mycotoxin biocontrol. While the literature on the use of *Penicillium* spp. as a biocontrol genus is not extensive, several systems were investigated. Examples include the use of *Penicillium nordicum* and *Penicillium chrysogenum* strains to mitigate the production of ochratoxin A (OTA) by *P. nordicum* in meat derivatives [21,22]. Avirulent strains of *P. chrysogenum* also demonstrated efficacy in controlling *P. expansum* in apples [23]. Additionally, various studies explored the use of *Penicillium* spp. as BCAs against other fungal species. For example, *Penicillium rubens* strain 212 and *Penicillium oxalicum* were employed as BCAs against *Fusarium oxysporum* f. sp*. lycopersici*, which causes vascular wilt in tomato plants [24]. *P. oxalicum* was found to be effective as a BCA in combating strawberry powdery mildew caused by *Podosphaera aphanis*, which is a primary fungal pathogen affecting this crop worldwide [25]. Furthermore, the application of microorganisms as protective cultures emerges as a promising strategy to prevent contamination from fungal diseases. Cebrián et al. [26] utilized *Debaryomyces hansenii* and *Staphylococcus xylosus*, which were isolated from dry-cured meat products, as BCAs against *Penicillium nordicum* during the maturation of sausages.

Bacteria and yeasts can also serve as alternative microbial agents in implementing this strategy. Specifically, yeast strains, such as *Aureobasidium pullulans* GE17, exhibit potent antagonistic effects against blue and green molds during in vitro growth. The mechanisms of this antagonism include competition for space and nutrients, secretion of lytic enzymes, release of volatile organic compounds, and inhibition of fungal spore germination [27]. This biocontrol strategy is also applicable to other species, including the use of non-aflatoxin-producing strains of *Aspergillus flavus*. Several commercially available products have been developed [28,29,30,31]. Biocontrol remedies generated worldwide using non-mycotoxigenic strains of *A. flavus* have a common characteristic: they use native *A. flavus* strains chosen from the specific region where they are intended to be applied [32]. The biocontrol potential of natural non-ochratoxigenic strains of *Aspergillus carbonarius* has also been recently described [33,34]. 

The objective of this work was to investigate the potential biocontrol activity of non-mycotoxigenic knockout mutants in mixed cultures with the wild type (WT). In this study, we aimed to compare the competitive behavior of two patulin-deficient mutants differing in secondary metabolite profiles. On one hand, we chose *patK*, which encodes a 6-methylsalicylic acid synthase, which is the initial enzyme in the patulin pathway, and, on the other hand, *veA*, which is a gene encoding a global regulator that is part of the VELVET complex. *P. expansum* Δ*patK* knockout mutants are expected to be affected only in patulin production, whereas Δ*veA* knockout mutants should be affected in different processes, including the production of a wider range of metabolites. 

## 2. Results

### 2.1. Generation and Characterization of ΔveA and ΔpatK Mutants 

To investigate the competitive dynamics between a non-mycotoxigenic and a mycotoxigenic strain, we used two different knockout mutants from the patulin-producing *P. expansum* strain CMP1. The first mutant involved the deletion of *patK* (∆*patK* mutant), which represents the first gene involved in the patulin biosynthesis pathway. This mutant was previously generated by Ballester et al. [12], who demonstrated that it is incapable of producing patulin during in vitro growth. The second mutant involved the deletion of *veA* (∆*veA* mutant), which is a global regulator of primary and secondary metabolism.

Figure 1 illustrates the gene substitution technique employed in this study. We utilized the Uracil-Specific Excision Reagent (USER)-friendly cloning method to create the gene replacement plasmid, namely, pRFHU2-veA (Appendix A) [35], which was introduced in *P. expansum* via *Agrobacterium tumefaciens*-mediated transformation. Transformants were analyzed using polymerase chain reaction (PCR) to select ∆*veA* mutants. Quantitative real-time PCR (qPCR) was conducted to determine the number of integrated T-DNA copies and to select knockout mutants without additional T-DNA integrations. An ectopic mutant containing a single copy of the deletion cassette in another region of the genome was also selected as a control, which could be compared with the WT strain.

At the macroscopic level, the WT strain exhibited a bluish-green appearance in the conidial region, accompanied by an outer white margin. In contrast, the Δ*veA* knockout mutant exhibited duller green without clear white borders (Figure 2A). These characteristics were observed in five biological replicates analyzed, indicating that the ectopic strain and the knockout mutants had stable phenotypes. The deletion of the *veA* gene resulted in abnormal conidiophores characterized by a sandy/velvety texture, a yellowish back colony color, and a characteristic earthy odor. Additionally, the surface of the knockout mutant colony appeared flat, unlike the other strains, where conidiophores clustered together, giving a grainy appearance, as described previously by El Hajj Assaf et al. [36]. Another notable difference was the inability of the null mutant to produce exudates [36]. Conversely, the phenotypic profile of the ∆*patK* knockout and ectopic mutants showed no significant differences when compared with the WT strain.

No statistically significant differences were observed in terms of growth or conidiation among the WT, ectopic strain, and ∆*patK* mutants (Figure 2B,C). Nevertheless, the growth and conidiation of the ∆*veA* knockout mutants were notably lower compared with those of the WT or the ectopic strain (Figure 2D,E). The conidiation of the ∆*veA* mutants was reduced by approximately 30% compared with that of the WT strain. Importantly, neither the ∆*patK* mutants, as described previously by Ballester et al. [12], nor the ∆*veA* knockout mutants (Figure 2F) were able to produce patulin during the testing conditions.

### 2.2. Effects of Different Stress Conditions on Growth of Knockout Mutants

To evaluate the possible phenotypic differences between parental and mutant strains that may affect the fitness, and thus, biocontrol efficacy, we analyzed their growth under different stress conditions. For this purpose, we incubated the strains in potato dextrose broth (PDB) supplemented with various stressor components at different concentrations (Figure 3 and Figure 4). The assay included different pH values and substances affecting the cell membrane and the cell wall, such as sodium dodecyl sulfate (SDS) and calcofluor white (CFW), respectively. In addition, the analysis included an examination of osmotic stress mediated by sodium salt (NaCl) or sorbitol, as well as the impact of oxidative stress imposed by hydrogen peroxide (H_2_O_2_).

The growth characteristics of the ∆*patK* knockout mutants were identical to those of the WT (Figure 3). Increasing concentrations of all tested stressor compounds, including H_2_O_2_, SDS, and NaCl, significantly influenced the relative growth inhibition (RGI) of both the WT and the Δ*patK* knockout mutants. Consistent growth patterns were observed only for varying concentrations of CFW and sorbitol. Both the WT and ∆*patK* knockout mutants exhibited reduced growth at pH 7.5. There were no significant differences in the growths of the WT and the ∆*patK* knockout mutants across any of the tested compounds.

Notably, the differences in growth between the ∆*veA* knockout mutants and the WT were more pronounced, as illustrated in Figure 4. Although the RGI observed in the presence of the highest concentration of H_2_O_2_ and SDS was similar for all strains, notable differences emerged in other scenarios, including growth in the absence of any stressor or the presence of either CFW, NaCl, sorbitol, or different pH values. In these cases, the deletion mutants exhibited considerably reduced growth. Specifically, in the growth medium supplemented with CFW at 70 µg/mL, it was observed that the ∆*veA* mutants showed better growth (lower % RGI), contrary to the situation at 2250 µg/mL, where higher relative growth of the WT was observed. Similar results were observed at high concentrations of sorbitol (1000 mM), where the growth of the deletant strains was lower (higher % RGI). Moreover, in the NaCl-supplemented medium, this differentiation in growth profiles, where the mutant strains presented a higher % RGI, was observed at both 70 mM and 2250 mM. Finally, a similar behavior was observed in the ∆*veA* knockout mutants across different pH values of 4.5, 6.0, and 7.5.

### 2.3. Effects of Gene Deletions on Competitiveness during In Vitro Growth

To investigate the competitive abilities of the knockout mutants compared with the WT strain, we conducted an experiment following the procedure outlined by Llobregat et al. [34]. In this experiment, we centrally co-inoculated the WT and one knockout mutant per construction (Δ*pks*-2 and Δ*veA*-6) at different ratios (10 WT:1 ∆, 1 WT:1 ∆, and 1 WT:10 ∆) on potato dextrose agar (PDA) plates. Additionally, we independently inoculated the WT and knockout mutants under the same conditions as the controls. Colony counts on PDA plates, with or without antibiotics, were conducted to confirm the initial proportion of each strain at the beginning of the experiment at time 0 (Figure 5A,B). It is important to note that only knockout strains were able to grow on PDA plates supplemented with antibiotics. Two independent experiments, with a minimum of three biological replicates in each experiment, were performed.

In this first experiment, colony counting (Figure 5C,D; solid bars) and qPCR (Figure 5E,F; solid bars) were used to determine the proportion of each strain under competition. According to the qPCR data, after 7 days of co-incubation of the WT and the knockout mutants inoculated at the 1:1 ratio, the percentage of the WT and ∆*patK* mutants remained relatively stable (48.2% WT:51.8% ∆*patK*) (Figure 5E). However, the ∆*veA* mutants were largely displaced by the WT (88.4% WT:11.6% ∆*veA*) (Figure 5F). As for the patulin production (Figure 6A,B, solid bars), when co-inoculating the WT and knockout mutant at an equal ratio (1 WT:1 ∆), the patulin levels decreased by 60% in the presence of the ∆*patK* knockout mutant compared with the control with only WT. However, when the ∆*veA* was co-inoculated with the WT at a 1:1 ratio, patulin levels were not reduced. A similar pattern was observed at the 1 WT:10 ∆ ratio. The expected decrease in patulin levels occurred when the WT was co-inoculated with the ∆*patK* mutant. (Figure 6A, solid bars). However, when the 1 WT:10 ∆ co-inoculation was done with the ∆*veA* mutant, the patulin level was identical to that observed when the WT strain was inoculated alone and much higher than the expected value.

In the second experiment, we aimed to determine the percentage of each strain at different time points: 0, 4, and 7 days of co-incubation. To calculate the proportion of each strain, we used colony counting at day 0 (Figure 5A,B; striped bars), the qPCR technique at day 4 (Appendix A, and Appendix A), and both colony counting (Figure 5C,D; striped bars) and qPCR at day 7 (Figure 5E,F; striped bars). In this second experiment, and based both on colony counting and qPCR data, the ratio of the WT and the ∆*patK* knockout mutant remained similar to the initial inoculation ratios at both 4 and 7 days post-inoculation. However, a significant displacement of the ∆*veA* knockout mutant by the WT strain was observed when they were co-inoculated at a 1:1 ratio (Figure 5D,F and Appendix A). A similar situation was observed for patulin production, with reductions of 48.4% and 23.6% for the ratios 1 WT:1 ∆*patK* and 1 WT:1 ∆*veA*, respectively, by day 4 (Appendix A). After 7 days of co-incubation of WT and the knockout mutants, the percentages of the WT and ∆*patK* were 66.8% WT:33.2% ∆*patK* when inoculated at similar proportions (1 WT:1 ∆) (Figure 5E,F). However, parallel to the results observed in the first experiment, the ∆*veA* knockout mutant was displaced by the WT at a co-inoculation ratio of 1:1, with percentages of 89.8% WT and 10.2% ∆*veA*. Furthermore, in the most favorable situation for the knockout mutants (1 WT:10 ∆), the ∆*patK* mutants maintained the initial co-inoculation ratio (11.9% WT:88.1% ∆*patK*), while the ∆*veA* mutant was displaced (77.3% WT:22.7% ∆*veA*). Regarding patulin production (Figure 6A,B, striped bars), under the most favorable conditions for the knockout mutants (1 WT:10 ∆), the ∆*patK* had reduced the patulin level by 91.6% by day 7. However, the reduction in patulin levels was only 25.1% in the co-inoculation of the WT and the ∆*veA*, when the expected reduction according to a 1 WT:10 ∆ ratio was 90.9%. 

## 3. Discussion

While promising biological control strategies have been developed for various *Aspergillus* species, as far as we know, the use of non-toxigenic strains as biocontrol agents for *Penicillium* sp. has not yet resulted in commercially available products. In this study, an analysis was conducted to assess the feasibility of using non-toxigenic knockout mutants of *P. expansum* to compete with a toxigenic WT strain of *P. expansum*. The aim was to decrease the population of the toxigenic strain, thereby reducing the overall patulin levels. We selected two different genes to test this hypothesis: *patK* and *veA*. On one hand, *patK* is responsible for encoding the 6-MSAS. The first step in patulin biosynthesis involves the formation of 6-MSA through the condensation of three malonyl-CoA units and one acetyl-CoA [17]. Its deletion causes a direct suppression of patulin production [12]. On the other hand, as described by El Hajj Assaf et al. [36], disruption of the *P. expansum veA* gene drastically reduces patulin production in vitro. The deletion of the *veA* gene in other fungi also leads to a reduction in other mycotoxins, such as OTA in *A. carbonarius* and *A. niger* [16,34,37], alternariol and alternariol monomethyl ether in *Alternaria alternata* [15], and aflatoxins in *Aspergillus flavus* [13,38]. Under our assay conditions, patulin production was also not detected in the Δ*veA* knockout mutants.

Regarding the potential impact of both *patK* and *veA* deletions on the phenotype, we analyzed the growth, conidiation, and stress responses, and we observed that the Δ*patK* knockout mutants were not affected in any of the analyzed conditions, which is consistent with previous findings by Ballester et al. [12]. However, the deletion of the *veA* gene resulted in phenotypic differences compared with the WT (Figure 2D,E). The Δ*veA* knockout mutants displayed altered conidiation and growth patterns, confirming previous results [36]. VeA appeared to be essential for patulin production, as the knockout mutants did not produce patulin when grown on PDA (Figure 2F). Numerous studies analyzed the involvement of VeA in fungal development, mainly in terms of sexual and asexual development. It is known that this protein plays a role in sexual development and acts as a negative regulator in asexual reproduction. According to the results of El Hajj Assaf et al. [36], the deletion of the *veA* gene caused a reduction in sporulation in *P. expansum* (Figure 2D). Similar results were observed also in *A. carbonarius* and *Fusarium fujikuroi* [34,37,39]. However, in *Aspergillus nidulans*, the *veA1* mutant sporulated better than the WT, especially under dark conditions [40]. It is important to take into consideration that *veA* gene expression is dependent on the growth media and the presence/absence of light [37,41]. Our findings regarding the impact of various stressors on the growth of the Δ*veA* knockout mutant (Figure 4) demonstrated a substantial detrimental effect caused by CFW, H_2_O_2_, NaCl, sorbitol, and different pH values.

During competitive assays, neither the ∆*patK* nor the ∆*veA* knockout mutants demonstrated the ability to outcompete the WT strain when initially introduced in an equal ratio (1 WT:1 ∆) (Figure 5C–F). Moreover, the ∆*veA* knockout mutant was almost entirely displaced by the WT in two independent experiments. The ∆*patK* knockout mutant exhibited a greater capability to maintain the initial co-inoculation ratio compared with the ∆*veA* mutant. Different results were observed using non-ochratoxigenic *A. carbonarius* strains. In this case, two different knockout mutants (Ac∆*pks*-Ac∆*otaA* and Ac∆*veA*) managed to displace the WT even under the worst-case initial inoculation conditions of 10:1 [34]. Notably, the Δ*patK* mutant did not alter the WT’s ability to produce patulin but influenced the population of the WT strain when both strains were co-inoculated, consequently affecting the overall levels of patulin. In a mixed inoculum containing both the WT and mutant strains, they coexist and eventually reach an equilibrium. The outcome depends on their respective fitness, leading to potential displacement toward the strain with higher fitness. Our observations indicate that a higher proportion of the non-patulin producer strain in the inoculum led to more substantial reductions in patulin production (Figure 6A), which parallels the decrease in the WT population. 

However, with the ∆*veA* knockout mutant, the patulin levels were always higher than expected according to the co-inoculation ratios, indicating the failure of this mutant to control the population of the WT strain and the subsequent production of patulin. The results presented by Llobregat et al. [34] indicated that the two non-toxigenic Ac∆*pks*-Ac∆*otaA* and Ac∆*veA* strains of *A. carbonarius* were able to displace the WT in co-inoculation experiments and significantly reduced the level of OTA content in the culture medium. The behavior of the Ac∆*veA* mutant contrasts with that of the Δ*veA* strain of *P. expansum*, where the fitness was lower than that of the WT. This fitness cost associated with the lack of the VeA regulator in *P. expansum* was not observed in *A. carbonarius*. Thus, removing the *veA* gene in both species gave rise to mutant strains with different competitive capabilities. On the other hand, the deletion of genes directly involved in the biosynthesis of the two mycotoxins led to mutant strains with improved competitive capabilities with respect to the WT strains, leading to a greater reduction in the corresponding mycotoxin in the culture medium as compared to mutants lacking the global regulatory *veA* gene. However, the Ac∆*otaA* mutant was able to displace the WT strain, whereas the *P. expansum* Δ*veA* mutant was not.

Nowadays, in the field of aflatoxin contamination in agricultural products, biocontrol using non-aflatoxigenic strains of *A. flavus* is considered the most effective option [42,43]. Recent studies showed that the use of the biopesticide Afla-Guard^®^ reduces aflatoxin levels produced by *A. flavus* in peanuts by 85.2% [44]. Similar results were obtained using the commercial product AF-X1^TM^, which is an atoxigenic *A. flavus* strain, reducing aflatoxin B1 levels in naturally contaminated maize fields by 92.3% [31]. Our results showed that when the WT and the Δ*patK* knockout mutant were co-inoculated under the most favorable conditions for the knockout, the percentage of patulin reduction averaged 94.4%, aligning with results achieved using commercial products. However, in the most favorable condition for the Δ*veA* knockout mutant, patulin levels were reduced only by 45.4%, while the anticipated reduction degree was 91%. This reinforces the idea that at least in *P. expansum*, knockout mutants in biosynthetic genes are the most promising candidates for a biocontrol strategy. However, the fact that the Δ*veA* mutant shows lower fitness than the WT strain might be useful to avoid the implementation of the mutant strain in the long term. A higher inoculum dose of the mutant could be necessary to obtain a practical reduction in the mycotoxin, but, on the other hand, its lower fitness would preclude its establishment in the environment. Future studies could further explore this possibility. 

In summary, our findings demonstrated a greater capability in reducing patulin levels when utilizing Δ*patK* knockout mutants. However, given VeA’s role as a global regulator of both primary and secondary metabolism, Δ*veA* mutants may also impact the production of other mycotoxins, such as citrinin. In alignment with strategies employed in *A. flavus* and *A. carbonarius* [31,34], there is potential to utilize non-mycotoxigenic strains of *P. expansum* to compete with toxigenic strains, leading to a reduction in patulin contamination in food and feed. This would be the first approach using the fungal species that is the main producer of blue mold disease, namely, *P. expansum*, to act as a biocontrol agent, but using atoxigenic strains. In addition, the development of knockout mutants opens the door to the possible complete suppression of the production of several mycotoxins, which is something that is not easily observed in natural atoxigenic strains. Moreover, some natural isolates defective in mycotoxin production contain an intact biosynthetic gene cluster, raising concerns regarding the possible synthesis of the mycotoxin under other environmental conditions [12].

## 4. Materials and Methods

### 4.1. Strains, Media, and Growth Conditions

The wild-type (WT) *Penicillium expansum* strain CMP1 (CECT 20906, deposited in the Spanish Type Culture Collection) was utilized in this study. All strains were grown on PDA (Difco-BD Diagnostics, Sparks, MD, USA) in the dark for 7 days, with or without the corresponding antibiotic. Conidia were collected from the agar using a sterile spatula, resuspended in distilled water, and quantified using a hemocytometer. *Escherichia coli* DH5α was used to propagate all plasmids and was cultured in Luria–Bertani medium (LB) supplemented with 25 µg/mL kanamycin at 37 °C. *Agrobacterium tumefaciens* AGL-1 strain used for fungal transformation was grown in LB medium supplemented with 75 µg/mL carbenicillin, 20 µg/mL rifampicin, and 100 µg/mL kanamycin at 28 °C.

### 4.2. Generation and Characterization of Mutant Strains

The Δ*veA* mutant strain was constructed using a homologous recombination strategy. Briefly, gene replacement was performed by employing the hygromycin resistance marker gene (*hph*) flanked by the 5′ upstream and 3′ downstream sequences of the *veA* coding region. All primer pairs were designed using the Primer3Plus version 3.3.0, following the same strategy described by Ballester et al. [12]. Primer pairs used to generate the null mutant in *veA* are indicated in Appendix A. The Δ*patK* mutant strain was previously obtained in a separate study [12]. Plasmid pRFHU2-veA (Appendix A) was obtained by cloning the disruption cassette in plasmid pRFHU2 [35] with the USER-friendly cloning technique (New England Biolabs GmbH, Frankfurt am Main, Germany), as previously described [45]. The 5′ upstream and 3′ downstream sequences of the *veA* coding region were mixed with the digested vector pRF-HU2 and USER enzyme mix (following the guidelines outlined by the manufacturer). Chemically competent *E. coli* DH5α cells were transformed using an aliquot of the USER mixture. PCR was employed to select the transformants using the primer pairs RF-5/RF-2 and RF-1/RF-6 (Appendix A). DNA sequencing was used to verify the correct fusion. 

The plasmid pRFHU2-veA was introduced into electrocompetent *A. tumefaciens* AGL-1 cells using the Gene Pulser apparatus (Bio-Rad, Hercules, CA, USA). The final transformation of *P. expansum* was performed as described previously [12]. An IMAS-induced culture of *A. tumefaciens* was combined in equal parts with a solution of *P. expansum* conidia (10^4^, 10^5^, and 10^6^ conidia/mL) and then inoculated onto paper filters placed on IMAS-containing agar plates. The membranes were transferred onto PDA plates supplemented with hygromycin B and cefotaxime after a co-culture at 24 °C for 48 h. After 3–4 days of incubation at 24 °C, *P. expansum* hygromycin-resistant colonies appeared. Transformants were confirmed via PCRs using genomic DNA isolated as described in López-Pérez et al. [46] (Appendix A and Figure 1).

Finally, qPCR was performed as previously described by Crespo-Sempere et al. [37] to determine the gene copy number of the T-DNA inserted in *P. expansum* using primer pairs PeveA-1F/PeveA-7R. The 37s ribosomal protein s24 gene and β-tubulin, which were amplified with primers Pe37S-F/Pe37s-R and Petub-1F/Petub-2R, respectively (Appendix A), were used as reference genes. qPCR reactions were carried out as previously described [47] using a LightCycler 480 Instrument (Roche Diagnostics, Mannheim, Germany) outfitted with the LightCycler SW 1.5 software. Calculations were done according to Pfaffl [48].

### 4.3. Macroscopic Morphology, Conidiation, and In Vitro Growth under Stress Conditions

From a suspension of 10^5^ conidia/mL, the WT and null mutant strains were inoculated with a central drop (5 µL) on PDA Petri dishes and grown for 7 days at 24 °C in the dark. Images were acquired daily, and the area of colonies was calculated using the ImageJ software version 1.53q. After the incubation period, acidic water was added to the plate and incubated with soft agitation for ten minutes. Total conidia were counted after scraping the plate with a spatula using a cell-counting chamber, and the acidic aqueous extracts were preserved at −80 °C for the assessment of patulin levels.

Growth profiles were obtained in 96-well PDB plates subjected to sequential concentrations of the compounds, as described previously [34]. Hydrogen peroxide (H_2_O_2_), sodium dodecyl sulfate (SDS), calcofluor white (CFW), sodium chloride (NaCl), sorbitol, and different pH values (3.0, 4.5, 6.0, and 7.5) were used to generate stress conditions. In triplicate, 96-well PDB plates, each containing 100 µL of the medium, were inoculated at a final concentration of 10^5^ conidia/mL. Plates were incubated at 24 °C for up to 7 days to analyze the growth profiles. Absorbance at 600 nm was automatically measured at 2 h intervals using a FLUOstar Omega (BMG Labtech, Ortenberg, Germany). Employing the spline fit model found in the grofit package version_1.1.1-1 [49], the software was capable of calculating four statistical parameters to quantify properties within each curve: length of the lag phase (λ), maximum growth rate (µ), maximum growth achieved (A), and area under the curve (AUC) at 7 days post-inoculation. Because the AUC is dependent upon the other three parameters, and thus, is more broadly discriminatory, we chose the comparison of AUC values. The inhibitory effect of different stresses on each strain was estimated using the relative growth inhibition (RGI), which was calculated as RGI (%) = 100 [(GC − GT)/GC], where GC (growth control) represents the AUC of each particular strain grown on PDB without stress or at pH 3.0, and GT (growth treatment) represents the AUC of fungus treated with different stress concentrations or increased pH in the 96-well assay. An RGI of 0 indicated no growth inhibition by the compound or compared with growth on PDB without stress or at pH 3.0, whereas an RGI of 100 corresponded to full growth inhibition. When comparing fungal growth in the absence of any stress, the control strain was the WT.

### 4.4. In Vitro Competition Assays

Competition assays using *P. expansum* CMP1 and non-mycotoxigenic mutants were performed under controlled laboratory conditions and in duplicate. A competition assay was performed on PDA plates using only one knockout mutant for each gene (Δ*veA*-6 and Δ*patK*-2). Five mixtures with a final concentration of 10^5^ conidia/mL at 10:0, 10:1, 1:1, 1:10, and 0:10 mix ratios (WT vs. Δ) were employed to inoculate 5 µL in the center of the PDA plates. Each mixing ratio was tested on five different culture plates. The experiment was repeated twice on different days. The plates were incubated for 7 days at 24 °C in the dark. After the competition assays concluded, the percentage growth of each competing strain and patulin production were assessed utilizing each plate. To determine the percentage of implantation of each strain and the presence of patulin, 3 mL of acidic water was incorporated into the plates, and the surfaces were lightly scraped. After 15 min of incubation, washing water was collected and used to analyze the presence of patulin using a high-pressure chromatography (HPLC) method, as previously described by Ballester et al. [12]. 

Two different approaches were performed to determine the growth percentage of each strain: (i) a suitable dilution of the collected acidic water–spore mixture was made to inoculate 100 µL of 2 × 10^3^ conidia/mL onto PDA plates and 200 µg/mL of hygromycin-supplemented PDA plates to count incipient colonies on the plate after 24–48 h (the ability to grow on PDA plates containing hygromycin was observed exclusively in the knockout mutants); (ii) the acidic water–spore mixture was centrifuged and the spore sediment was recovered to extract the DNA and to quantify the percentage of growth of each strain using qPCR. Using specific primers, the DNAs of *P. expansum* CMP1 and the mutants were utilized to determine the growth percentage of each strain; PeveA-5F/PeveA-6R and PepatK-F/PepatK-R were used to identify the *veA* and *patK* genes in the WT strain, respectively; HPH3F/HPH4R was used to identify the hygromycin-resistant marker in the knockout mutants, and Pe37s-F/Pe37S-R was used to identify the reference gene, which was present in all the strains. qPCR was performed as described in the previous section.

### 4.5. Statistical Analysis

Statistical analyses were performed using statgraphics stratus (Statgraphics Technologies, Inc., The Plains, VA, USA). Levene’s test was used to determine the homogeneity of the variances. An analysis of variance (ANOVA) was performed to determine the existence of significant differences between samples, followed by a Tukey’s HSD post hoc test (*p* < 0.05). All error bars in all graphs were generated using the standard error (SEM). 

## Figures and Tables

**Figure 1 toxins-16-00052-f001:**
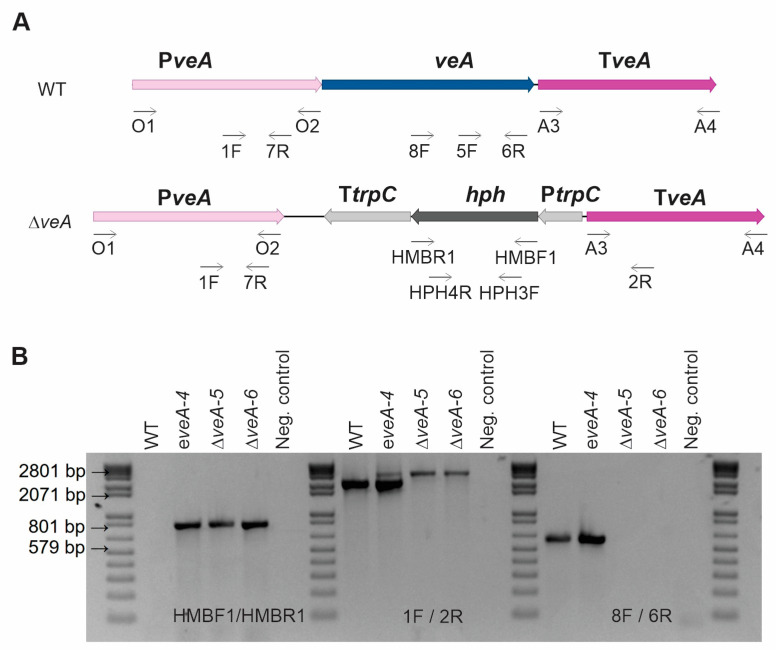
Construction of the *veA* mutants and molecular analysis. (**A**) Schematic diagram of the deletion cassette used to replace the target region (gene of interest) in the wild-type (WT) strain using homologous recombination, generating Δ*veA* mutants. Primers used in the construction are indicated. (**B**) Ectopic and knockout mutants were tested using PCR analysis with genomic DNA as template: amplification to detect the selection marker (hygromycin) in the mutants in the left panel; amplification to detect the presence and deletion of the gene of interest (*veA* gene) in the middle panel (PeveA-1F/PeveA-2R; Appendix A); and amplification to detect the gene of interest (*veA* gene) in the WT and the ectopic strains, giving a negative result in the knockout mutant (PeveA-8F/PeveA-6R), in the right panel. Analyses were performed for WT, one ectopic mutant (indicated by “e”), and two knockout mutants (indicated by “Δ”).

**Figure 2 toxins-16-00052-f002:**
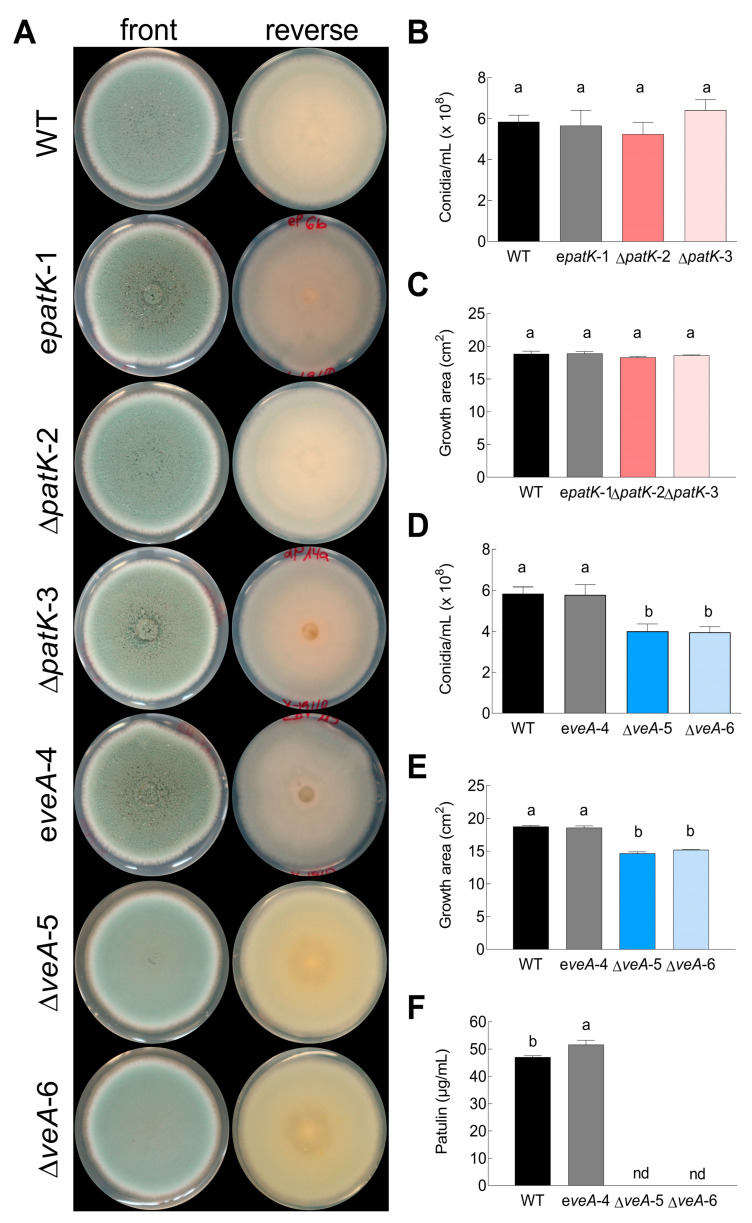
Morphological phenotype of *P. expansum* CMP1 (designated as “WT”, black bars), an ectopic mutant (designated as “e”, gray bars), and two knockout mutants of *patK* (**B**,**C**; denoted as ∆*patK*) and *veA* (**D**–**F**; denoted as ∆*veA*). (**A**) Comparison of the phenotypes of the different strains grown on PDA plates. The images are representative of five independent replicates. Conidiation (**B**,**D**), growth area (**C**,**E**), and patulin production (**F**) were assessed on PDA plates. Data represent the mean ± standard error of the mean (SEM) from at least three biological replicates. Letters show significant differences among samples (one-way ANOVA and Tukey’s HSD test, *p* < 0.05). “nd” denotes non-detected during the testing conditions.

**Figure 3 toxins-16-00052-f003:**
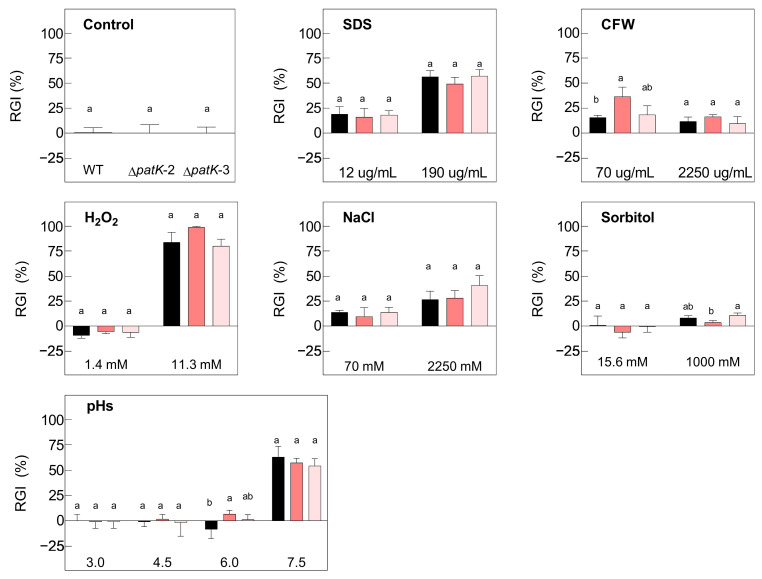
Relative growth inhibition (RGI) of the WT *P. expansum* (black bars) and two ∆*patK* knockout mutants (colored bars) in the absence of stress (control graph) and in the presence of different concentrations of H_2_O_2_, CFW, SDS, NaCl, and sorbitol, and at different pH values. The area under the curve (AUC) was utilized to determine the growth 7 days post-inoculation at 24 °C. Different letters show statistical significance between the knockout mutants and the WT (one-way ANOVA and Tukey’s HSD test, *p* < 0.05). Bars show the mean ± standard error of the mean from three biological replicates and are representative of two independent experiments.

**Figure 4 toxins-16-00052-f004:**
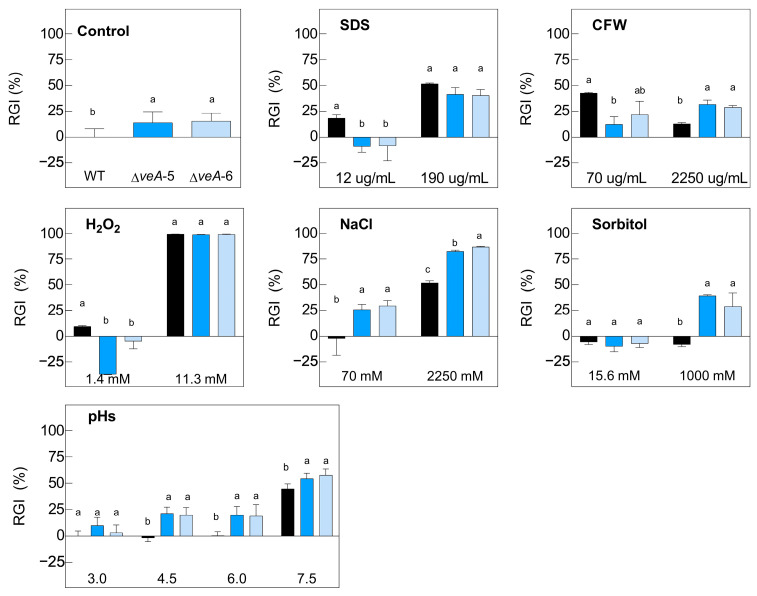
Relative growth inhibition (RGI) of WT *P. expansum* (black bars) and two ∆*veA* knockout mutants (colored bars) in the absence of stress (control graph) and under different concentrations of H_2_O_2_, CFW, SDS, NaCl, and sorbitol, as well as at distinct pH values. The area under the curve (AUC) was used to determine the fungal growth 7 days post-inoculation at 24 °C. AUC was compared using one-way ANOVA and Tukey’s HSD test (*p* < 0.05) to establish whether significant differences (indicated by different letters) existed between the knockout mutants and WT. Data shown are mean ± standard error of the mean of three biological replicates and are representative of two independent experiments.

**Figure 5 toxins-16-00052-f005:**
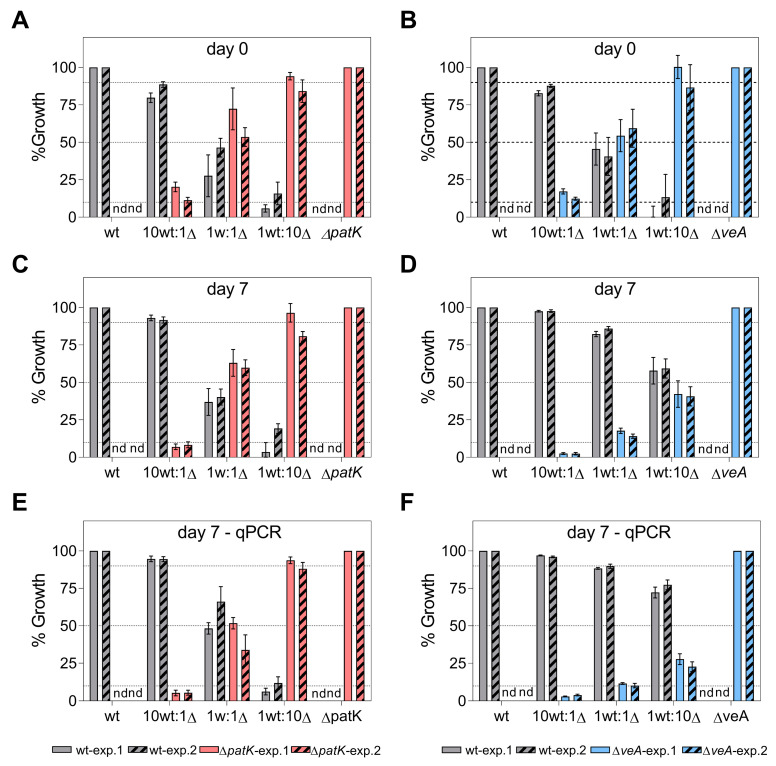
The competitiveness of Δ*patK* (**A**,**C**,**E**, red bars) and Δ*veA* (**B**,**D**,**F**, blue bars) knockout mutants against the patulin producer *P. expansum* CMP1 (“wt”, grey bars) was assessed at day 0 (**A**,**B**) and 7 days post-inoculation (**C**–**F**) on PDA plates. Competition assays were conducted at ratios of 10:0, 10:1, 1:1, 1:10, and 0:10 (WT vs. ∆). Estimations of each strain’s percentage were made by counting colonies on both PDA and hygromycin-supplemented PDA (**A**–**D**) and using qPCR (**E**,**F**). Growth on PDA supplemented with the antibiotic was observed exclusively in the knockout mutants. The analysis was repeated in two independent experiments (solid bars and stripped bars for the first and the second experiment, respectively), with at least three biological replicates in each experiment. Bars represent the mean values and the standard error of the mean from at least three biological replicates for each experiment. The expected values for the WT at the different co-inoculation ratios are indicated by the dotted lines. Statistical analyses are presented in Appendix A. “nd” denotes not detected during the testing conditions.

**Figure 6 toxins-16-00052-f006:**
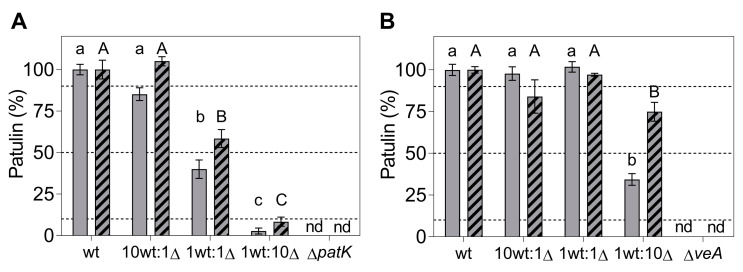
Patulin production at 7 days post-inoculation during competition assays of WT vs. ∆*patK* (**A**) and ∆*veA* (**B**) knockout mutants. The competition assays were conducted at ratios of 10:0, 10:1, 1:1, 1:10, and 0:10 (WT vs. ∆) on PDA. Two independent experiments are presented, indicated by solid and striped bars, respectively. Each experiment involved the analysis of at least three independent biological replicates. The dotted lines indicate the expected values for the WT at the different co-inoculation ratios. Data represent the mean ± the standard error of the mean of at least three biological replicates for each experiment. Letters in the same panel show significant differences for the same experiment (one-way ANOVA and Tukey’s HSD test, *p* < 0.05). “nd” denotes not detected during the testing conditions.

## Data Availability

The data presented in this study are available in DIGITAL.CSIC (http://hdl.handle.net/10261/342448).

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
