# Peer review of "Exploring the Biocontrol Capability of Non-Mycotoxigenic Strains of Penicillium expansum"

_toxins, 2024, doi:10.3390/toxins16010052_

Round 1

Reviewer 1 Report (Previous Reviewer 1)

Comments and Suggestions for Authors

I was interested to see the corrections made by the authors.

Unfortunately, some of them, highlighting some previously questionable issues, now further confirmed my belief that the direction and thoughts of the authors need to be re-edited.

It seems to me, however, that in order not to waste such a solid work, it is definitely necessary to change the purpose and intentions that the authors have shown in their work.

It just doesn't seem to me, from an image point of view, that fighting with a knocked out strain against a wild strain is applicable. All the more,  because the whole concept of biocontrol makes sense only when the pathogen ceases to exist and it can be seen, and in this case it will not be seen.

Author Response

We apologize that the strategy of utilizing knockout mutants as biocontrol agents does not align with the reviewer´s preferences. The methodology we have employed serves as a proof of concept, just to check whether non-mycotoxigenic strains could reduce the levels of patulin in coculture experiments. We are aware that there is still progress to be made before these strains can be developed as a biocontrol agent in the field. The use of Agrobacterium tumefaciens-mediated transformation generates genetically modified microorganisms. Nevertheless, employing genetic editing via CRISPR-Cas9 provides a viable alternative. For instance, the application of CRISPR-Cas9 facilitates the deletion of three mycotoxin biosynthesis gene clusters in Aspergillus flavus, as it has been described recently (Creating large chromosomal segment deletions in Aspergillus flavus by a dual CRISPR/Cas9 system: Deletion of gene clusters for production of aflatoxin, cyclopiazonic acid, and ustiloxin B (https://doi.org/10.1016/j.fgb.2023.103863). The authors indicate that “The research provides a method for creating genuine atoxigenic biocontrol strains friendly for field trial release." In our laboratory, we are in the process of generating knockout mutants genetically manipulated through CRISPR-Cas9. 

Generating mutants through genetic engineering enables the development of completely nontoxigenic mutants, which do not produce any of the mycotoxins that can be produced by the parental strain. Furthermore, the CRISPR-Cas9 technique leaves no traceability, as no foreign DNA is integrated into the genome, and in several countries, these gene-edited microorganisms are no longer considered genetically modified organisms. This fact paves the way for the development of non-mycotoxigenic biocontrol microorganisms that are not able to produce any mycotoxin. 

Reviewer 2 Report (Previous Reviewer 2)

Comments and Suggestions for Authors

The authors fully answered to the previous remarks, the manuscript can be published

Author Response

We thank the reviewer for previous corrections that have allowed us to improve the manuscript. 

Reviewer 3 Report (Previous Reviewer 3)

Comments and Suggestions for Authors

I found the manuscript well implemented with respect to the previous version I received. As well, Authors convincingly replied to my comments and followed my suggestions.

I only have one more, that could bring a little additional value and support the really important need of choosing (and validating) the fungal biocompetitors within the ecological niche of the resident microbial population:

Line 129-130: in order to better discuss the central topic of statement "These atoxigenic A. flavus-based biocontrol solutions produced worldwide share the characteristic of employing native A. flavus strains selected from the region where they will be used", I suggest a very relevant publication by Spadola et al.: "Validation and Ecological Niche Investigation of a New Fungal Intraspecific Competitor as a Biocontrol Agent for the Sustainable Containment of Aflatoxins on Maize Fields." (J. Fungi 2022, 8, 425. https://doi.org/ 10.3390/jof8050425). 

Author Response

We greatly appreciate the reviewer's comment and regret not having come across this article earlier. We have found the suggestion very interesting and helpful for our discussion. Consequently, we have included the reference in the manuscript. 

Round 2

Reviewer 1 Report (Previous Reviewer 1)

Comments and Suggestions for Authors

The revisions have enhanced the publication and I wish the authors much success in their continued research.

This manuscript is a resubmission of an earlier submission. The following is a list of the peer review reports and author responses from that submission.

Round 1

Reviewer 1 Report

Comments and Suggestions for Authors

Given the broad introduction to BCAs, although definitely not very cross-sectional, the first question that comes to mind is why create a strain with a knocked out gene instead of fighting it.

Despite this, the authors quickly showed that any genetic interference affects the phenotypic response of the organism. This is particularly evident in RGI systems.

In addition, it is worth noting that the authors themselves have indicated that in co-cultures, not only the wild strain is not eliminated, but even stimulated to grow and biosynthesize patulin.

Thereby. While it is difficult to accuse the paper of interesting and logical experimentation, at the end, the question of where is the goal that the authors wanted to achieve or indicate?

Maybe it is enough to rewrite the discussions in a different approach, or partially supplement them with some experiment that will give a more optimistic tone to the publication.

Reviewer 2 Report

Comments and Suggestions for Authors

“Exploring the biocontrol capability of non-mycotoxigenic strains of Penicillium expansum” is the title of the proposed article. The authors analyzed two knockout mutant strains (∆patK and ∆veA). Growth, morphological and biochemical analyzes under normal and stress conditions were described between the mutants and the WT. The results highlighted differences between the two mutants compared to the WT, indicating the ∆patK mutant as the best performer for the purposes of the work.

The research has been carried out properly, the manuscript is clearly written, and the results are of interest. Some little revision as follow, and the manuscript can be published.

Points that need to be addressed:

Figure 1: a) please, improve image quality; b) the authors should enlarge panel B (at least the size of panel C), otherwise the name and orientation of primers are almost invisible.

Line 189: please, the authors must specify that the results shown in panel 2A are not coming from a single growth, especially for the mutants. In short, make it clear that it is a stable phenotype.

Figure 2: a) please, improve image quality; b) please, authors should enlarge panels B-F. Axis dimensions and column names are barely readable. Improve photo quality.

Section 2.2: please, the authors should write a clearer sentence at the beginning of the section to indicate the motivation for doing experiments under stress conditions.

Figure 5: please, improve image quality and panels’ size

Figure 6: please, improve image quality and panels’ size and if I understand you correctly, perhaps it is better to put panels A and B of figure 6 in figure 5 and then leave panels C and D in figure 6. It is not fundamental, just to have a more streamlined reading.

Reviewer 3 Report

Comments and Suggestions for Authors

The exploitation of atoxigenic strains for the biocontrol of mycotoxigenic fungal species has been long time investigated; over decades, various attempts and experimentations have been done to understand the biological mechanisms ruling the intra- and interspecific inhibition of mycotoxins biosynthesis determined by the co-occurrence of non-producing isolates, clarifying that the mere competitive exclusion from the niche is not the driving force of such phenomenon. Here, a study involving two knock-out strains of P. expansum is presented, that try to demonstrate that in this species also a biocompetitve strategy using atoxigenic strains might by promising for the containment of patuli in food commodities.

Despite the topic is of high interest to me, there are major issues I would like to raise that unfortunately make the manuscript not acceptable in the present form.

1) Introduction:

I found this section a little bit unbalanced: for example, while huge attention has been paid to the fungus and the stategies used that employ BCA application, no description at all has been provided to the patulin biosynthesis and the environmental/developmental conditions involved in the mycotoxin accumulation (that is fundamental to unravel the intraspecific biocontrol mechanisms). Neither mention about the patulin gene cluster is present, nor any information about its precursors, intermediates, pathway enzymes.... Author MUST re-think the introduction in this perspective, in order to give the Readers the most coherent background.

2) Results: a lot of assay have been performed, and I must say that they are well described and presented. However, two experiments are not enough when it comes to this kind of analysis, that are simple and low costing. At least three independent repetition must be performed. Additionally, statistical analysis should be  

3) Discussion:

Despite the amount and the different nature of data presented, Authors failed to critically discuss their findings. As a result, the section seems scarce, being almost a simple repetition of Intro and Results: for example, lines 334-356 are redundant with concepts already provided in the Introduction, and don't belong to the discussion (hence, delete this paragraph!). On the other hand, the biological significance of obtained results is not clear at all: essential questions such as why deleting the first gene of the patulin pathway could affect, in the toxigenic wt strain, the biosynthesis of mycotoxin, why this specific gene was choosen, or why the depletion of patK was expected to alter the response of P. expansum to osmotic or salt stress, have not been formulated nor answered. In addition, given VeA’s role as a global regulator of BOTH primary and secondary metabolism, why Authors thought that the use of ΔveA mutants should provide evidence of any intraspecific biocompetition mechanism against patulin accumulation? What is the original contribute to the state of the art of this experimentation?

As follows, there are also a few minor points requiring attention:

Line 89-91: Authors stated that "there are no previous studies on the competitive ability of non-mycotoxigenic Penicillium strains as biocontrol agents (BCAs)", but it's nor true: please cite "Ochratoxin A control in meat derivatives: Intraspecific biocompetition between Penicillium nordicum strains [Berni, E.; Montagna, I.; Restivo, F. M.; Degola, F. JOURNAL OF FOOD QUALITY. - ISSN 0146-9428. - 2017:(2017), pp. 1-8. [10.1155/2017/8370106]

Line 117-127: this paragraph is well described, but it distracts the Reader and should be shortened. Please reduce to: "This biocontrol strategy is also applicable to other species, including the use of non-aflatoxin-producing strains of Aspergillus flavus. Several commercially available products have been developed [20, 21, 23, 24, 26]. These atoxigenic A. flavus-based biocontrol solutions produced worldwide share the characteristic of employing native strains selected within the region where they will be used. The biocontrol potential of natural non-ochratoxigenic strains of Aspergillus carbonarius has also been recently described [26, 27]" and delete, accordingly, references 22, 25

Line 130-135: "These specific strains were chosen....biocontrol agents [27]" is redundant; please delete.

Figure 2A: please amend the figure changing "back" with "reverse", that is the correct word for the Petri dish bottom-up view.